# A realist evaluation of the development, implementation and outcomes of the first public ART Centre in Morocco

**Amal Benbella**[1,2]*, **Asmaa Zaidouni**[2], **Sanae Elomrani**[3], **Gitau Mburu**[4],
**Houyam Hardizi**[1,2‡], **Abdelhakim Yahyane**[5‡], **Hafid Hachri**[6‡], **Karima Gholbzouri**[7],
**James Kiarie**[4], **Rachid Bezad**[1,2]

1 Faculty of Medicine and Pharmacy, Mohammed V University, Rabat, Morocco, 2 Maternity and Reproductive Health Hospital of IbnSina University Hospital, Rabat, Morocco, 3 Faculty of Sciences, Biology and Health Laboratory, Ibn Toufail University, Kenitra, Morocco, 4 Department of Sexual and Reproductive Health, Maternal, Child, Adolescent Health and Ageing, Human Reproduction Program, World Health Organization, Geneva, Switzerland, 5 Population Directorate, Ministry of Health and Social Protection, Rabat, Morocco, 6 World Health Organization Country Office, Rabat, Morocco, 7 Reproductive Health and Research Unit, WHO Eastern Mediterranean Regional Office, Cairo, Egypt

☙ These authors contributed equally to this work.
☙ These authors contributed equally to this work.
‡ HH, AY and HH also contributed equally to this work.
* amalben.25@gmail.com

## Abstract

In Morocco, access to assisted reproductive technology (ART) in public hospitals has been limited, and many couples cannot afford private sector services. In this context, a public ART centre was established in 2013 at the Maternity and Reproductive Health Hospital in Rabat. This study aimed to evaluate this first public ART centre to understand how the intervention worked, for whom, and under what circumstances, and to inform replication in Morocco and other low- and middle-income countries (LMICs). A realist evaluation was conducted, combining qualitative in-depth interviews with couples, healthcare providers, and policymakers, and a retrospective analysis of clinical outcomes among all couples receiving ART services at the centre. Data were analysed to develop context–mechanism–outcome configurations explaining how and why the intervention worked. Contextual factors such as the high cost of IVF, the social value of childbearing, and stigma associated with infertility influenced service utilization. Key mechanisms included advanced planning, workforce training, standardized procedures, patient-centred care, and strong leadership and governance. Outcomes included the establishment of a regulatory framework for ART, perceived improvement in financial affordability, and perceived reduction in infertility-related stigma. Knowledge transfer contributed to the development of additional centres in various regions of the country. Between October 2013 and December 2022, 2,495 couples received fertility services. Clinical pregnancy rates were 23.8% for IVF and 21% for frozen embryo transfer, with corresponding live birth rates of 16.6% and 14.3%. The creation of the first public ART centre represents a milestone

**Data availability statement:** All participants provided informed consent for their interview transcripts to be published. The transcripts have been fully de-identified to remove any potentially identifying information, including names, initials, addresses, or specific dates. Anonymized data underlying the findings described in this study are available in the Supporting Information files (S1 Data and S2 Data).

**Funding:** This work was supported by the UNDP–UNFPA–UNICEF–WHO–World Bank Special Programme of Research, Development and Research Training in Human Reproduction (HRP), a co-sponsored programme executed by the World Health Organization. Four co-authors (GM, HH, KG, JK) are staff members of the World Health Organization. The views expressed in this article are those of the authors and do not necessarily represent the views, decisions, or policies of the World Health Organization. The funders had no additional role in data collection and analysis, or the decision to publish.

**Competing interests:** GM, HH, KG, and JK are staff members of the World Health Organization. The authors declare no other competing interests.

in the management of infertility in the Moroccan public health system. It has improved access and perceived affordability to ART, although some clinical outcomes require further improvement. These findings provide useful lessons to inform the development of public ART services in other LMICs.

## Introduction

Infertility is a disease of the reproductive system defined by the inability to achieve clinical pregnancy after 12 months or more of regular unprotected sexual intercourse [1]. According to the World Health Organization's 2023 estimates, approximately 17.5% of people globally experience infertility at some point in their lives [2]. Infertility has serious impacts on mental well-being [3], is a source of multidimensional stress with negative social consequences [4–6], and predisposes infertile couples to stigmatization, social isolation and loss of status [6]. In addition to its impact on mental and social wellbeing, infertility is also a source of financial burden [7]. Apart from its links on health and economics, infertility is also linked to human rights of those involved [8,9]. It limits individual ability to become a parent, secure the family line [10] and achieve individual fertility preferences [11]. Thus, infertility represents a common concern for public health professionals and decision-makers [12].

Despite the burden of infertility globally, infertility is not prioritized in many low and middle income countries (LMICs) [12,13], and provision of infertility services is predominantly in the private sector, resulting in limited information about set up of public infertility services in these settings. In many LMICs, where fertility rates are high, infertility is not viewed as a pressing problem [14], which limits investments aimed at enabling couples to reproduce (e.g., through ART), compared to those directed at reducing fertility rates [14,15]. Although there is international consensus on the concept of reproductive health, and on the need to safeguard the rights, freedom, and ability of individuals to decide the number, timing and spacing of their children [16], infertility has not been a major focus in the global sexual and reproductive health and rights (SRHR) agenda [9], despite the link between infertility and Sustainable Development (SDG) Goal 3 (to ensure healthy lives and promote wellbeing for all at all ages) [17].

The limited global political commitment and interest, coupled with the high cost of ART, has resulted in considerable inequity in the availability and accessibility of fertility care worldwide [9]. Significant gaps persist in the provision of most infertility services [9], fertility education is far from universal, and monitoring of fertility care is inconsistent [9]. Governmental investments in service delivery, health workforce, health information systems, essential medicines, financing, leadership and governance of fertility are generally inadequate in many countries. Yet these building blocks have been identified by the World Health Organization (WHO) as essential in achieving universal health coverage [18].

In Morocco, a 2015 nationwide survey found that 12% of couples had difficulty conceiving [19]. The 2018 National Population and Family Health Survey also found

a shift in reproductive patterns, with declining total fertility rates and rising maternal ages [20]. These data underline the relevance of infertilityservices in Morocco.

However, despite the relevance and the recognition that access to reproductive healthcare is a right for every Moroccan citizen, addressing infertility had not been prioritized in previous decades. Morocco has had successive policies on health, such as the Health Action Plan 2008–2012 [21] and the Health Sector Strategy 2012–2016 [22], which focused on maternal and neonatal mortality and contraception; however, advocacy and political leadership around infertility lagged behind [16]. It was not until 2025 that ART was added in the National Health Plan, Axis 13, with a call to "Launch new health programs and strategies", and within it, Action 59 was to launch the "National plan for ART" [23]. Given this policy objective, the creation of an ART centre incorporating infertility services in the public sector was proposed [24]. The public ART centre was conceptualized in 2008, and became operational in 2013.

### Study aim

Given that this was the first public ART centre of its kind in Morocco, and in-order to disseminate this experience at the national and international levels, we conducted a study to evaluate its development, implementation and outcomes in-order to enhance the understanding of how such initiatives can be replicated in other LMICs.

## Materials and methods

### Study design

This was an evaluation study which is suitable for the stated study aim, incorporating qualitative exploration and retrospective data analysis. The reporting of qualitative results is in keeping with the consolidated criteria for reporting qualitative research (COREQ) requirements [25]. (Please see S1 Table).

### Study setting

The study was conducted at the ART centre of the Maternity and Reproductive Health Hospital (HMSRO) of Ibn Sina University Hospital (CHUIS) in Rabat. The ART centre is the first public fertility centre in Morocco and receives couples from different regions of the country. In addition, it is a referral centre catering for many couples referred for ART.

### Setting up of the centre

The ART centre was established through technical collaboration with Belgian partners, namely the Belgian Academy of Research and Higher Education, the Free University of Brussels, Erasme Hospital, University of Liege and La Citadelle Hospital. Several components were directly replicated from these partners, including core laboratory protocols, infection prevention and control standards, equipment specifications, quality control procedures, and embryology laboratory layout. In addition, Gyneco 2000 software was also brought in from Erasme Hospital where it had been implemented in their ART centre.

However, multiple adaptations were necessary to ensure alignment with Moroccan sociocultural, legal, and health system realities. Eligibility criteria were restricted to married couples, in accordance with Moroccan law and religious principles. Counselling protocols were adapted to address infertility-related stigma, gender dynamics and religious considerations prohibiting the use of third-party gametes. Treatment regimens were adapted to match the availability of registered medications and consumables. These adaptations allowed the model to be both technically sound and culturally acceptable.

The centre's implementation relied on building a skilled, multidisciplinary team through targeted capacity building. The cadre mix included reproductive gynaecologists, reproductive biologists, laboratory technicians, and nurses. They received training and continued professional development in infertility diagnosis, ART procedures, and laboratory techniques.

The ART centre started treating couples in 2014 and was publicly inaugurated and formally recognized by government officials in May 2016. Although services had begun prior to the inauguration — with trained personnel, established protocols, and all necessary supplies — regulatory and logistical foundations were not fully in place until May 2016, and it required further efforts to ensure that the centre operated optimally.

### Nature of services provided at the ART centre

The ART centre offers a broad range of services designed to address infertility in couples. These services include initial diagnostic assessments, hormonal evaluations, ultrasound monitoring, and semen analysis. The centre provides ovulation induction, intrauterine insemination (IUI), in vitro fertilization (IVF) and cryopreservation of embryos. Diagnostic and therapeutic procedures such as hysteroscopy are performed when indicated, and in selected cases, laparoscopic surgery is provided to address underlying gynaecological conditions affecting fertility. Nurse-led consultations support patients throughout the care continuum, offering education, treatment monitoring, and counselling on self-care practices including medication administration, lifestyle modifications, and stress reduction. Psychological consultations are also available to help individuals and couples cope with the emotional and relational challenges associated with infertility and its treatment. Couples care pathway is shown in S1 Fig.

### Populations served by the ART centre

From October 2013 to December 2022, 2495 couples were provided with different services. The couples were of reproductive age and many will have had infertility for a few years (average 6 years, range of 1–27 years) before contacting our centre. In general, half of the infertility cases seen at the centre were due to a male factor while the remaining tended to be due to a female factor. The couples at the centre came from all regions of the country. Few (15 cases (0.6%) travelled from foreign countries to obtain services at the centre. Generally, the ART centre serves a wide array of households. Two thirds of female patients had been housewives and roughly a third of men had been employed. Just over half (52.9%) of the patients had consulted another health provider before contacting our centre.

### Organization, governance and financing of care at the ART centre

At the centre, both male and female partners undergo diagnosis and treatment when indicated. Medical and nursing consultations are routinely conducted with both partners present, through a couple-centred approach. Even in cases where the infertility cause was female, male partners are actively involved in counselling, treatment decision-making, and follow-up. (Please see S2 Table).

Management of couples is facilitated by the use of Gyneco 2000 software which enables centralized patient medical information and treatment history, monitoring of ovarian stimulation and treatment protocols, and tracking of gametes collections and embryo transfers. The health information system integrates biological and imaging results and generates personalized reports. It allows a gender-disaggregated data entry, the monitoring of male and female infertility cases separately, and clinical and programmatic decision-making, while ensuring confidentiality and maintaining secure backups of medical records.

As there is currently no insurance coverage or state subsidy for ART in Morocco, the centre charges user fees; and the payment system operates on a bundled payment model, covering the entire treatment cycle. Cost transparency measures are implemented to ensure couples receive clear information about standardized fees prior to treatment initiation, facilitating informed consent and decision-making.

The governance of the ART centre is anchored within CHUIS, and Mohammed V University with the centre is physically located at the HMSRO, with the Ministry of Health providing policy and regulatory oversight. Decision-making is coordinated through regular meetings involving clinical, laboratory, nursing, and administrative leaders, enabling planning,

problem-solving, and effective policy implementation. Established accountability mechanisms include periodic performance reviews, financial reporting, and service delivery monitoring. Institutional coordination is extended to other CHUIS departments such as pharmacy, biomedical services, and the central laboratory, as well as external partners. (Please see S3 Table for a detailed timeline of the implementation of the ART centre).

## Evaluation rationale and approach

When any program is implemented, it is testing a theory about what and for whom the intervention might work, and under what circumstances, and specifically what 'might cause change' even though that theory may not be explicit when the program is set up [26]. When the Public ART centre was initiated in Morocco, the theories or mechanisms by which change might occur in increasing access to ART were not explicit. Therefore, a key task of our realist evaluation was to make the theories within the program explicit, by developing clear hypotheses about how, and for whom, programs might 'work' and why. We employed the approach proposed by Pawson and Tilley [27], that focuses on Context, Mechanisms and Outcomes, and further elucidated by de Weger et al [28] (Table 1). Given that realist evaluations are method neutral, the combination of different methodologies was chosen as it would enhance the elaboration of how the fertility care may work, using triangulation of different data.

## Participant recruitment

To understand the context and mechanisms of the intervention, 39 participants were selected using purposive sampling from three different groups to ensure representation and diversity of perspectives.These three groups were: (i) couples, (ii) health care providers and (iii) policy stakeholders who have been involved in setting up the centre,as shown in the following Table 2.

## Informed consent procedures

Written informed consent was obtained from each participant. Prior to consenting, participants were given explanations on the purpose of the study, the researchers' role, and the intended use of the findings, and were reassured that their information would be confidential. Participants were informed that they don't have to participate in this research and that even if they decide

**Table 1. Definitions of Key Realist Concepts, adapted from de Weger et al [28].**

| Concept | Definition |
|---|---|
| Context | Refers to the background of a program, including pre-existing socio-economic, political and institutional policies and structures, socio-cultural norms, and their interrelationships. Some aspects of contexts may either facilitate or prevent some mechanisms to be triggered. |
| Mechanism | Mechanisms describe how the resources embedded within a program influence the reasoning and behaviour of program participants. Mechanisms are usually hidden, sensitive to variations in context, and generate outcomes. |
| Outcome | Refers to intended, unintended, or unexpected program outcomes on the micro-, meso-, or macro-levels |
| Context-Mechanism-Outcome (CMO) | CMO is a term used to explain generative causation, reflecting on the relationship between a context,mechanism, and an outcome of interest in a particular program. Configuring CMOs is a basis for generating or refining (program) theories. |

**Table 2. Study Participants.**

| Participant type | Sample sizes |
|---|---|
| Couples receiving fertility care at the centre | • 6 couples who have completed treatment (n = 12)<br>• 4 couples who are in the process of receiving IVF (n = 8)<br>• 4 couples who are on the waiting list for IVF (n = 8) |
| Providers of fertility care | • Medical doctor (n = 2)<br>• Midwife (n = 1)<br>• Laboratory technician (n = 1)<br>• Healthcare provider from referring hospital (n = 2) |
| Policy stakeholders involved in developing the fertility care services | • ART Centre Director (n = 1)<br>• Ministry of Health (n = 2)<br>• Public Health expert (n = 1)<br>• Professional Fertility Association (n = 1) |

to participate, they will have the right to refuse to answer any questions, or to revoke their consent and cease participation at any time they wish without giving any reasons, without any consequences whatsoever. All individuals who were invited to participate in the study agreed to take part and provided informed consent. There were no refusals or withdrawals after consenting.

## Data collection

Two researchers (SE and AZ), both with training and prior experience in conducting qualitative interviews, carried out the data collection. SE is a public health researcher with qualitative research experience in health systems evaluation, and AZ is a health professional with additional training in qualitative methods. Both were familiar with realist evaluation methodology. The researcher met the study participants at the time and place agreed by both parties and where privacy and confidentiality could be ensured (at the ART centre and at relevant health administration offices). Permission was sought to record the interview on an audio device and to take notes by hand.A semi-structured interview guide for each participant group was used to elicit the perspectives and experiences of participants. Each guide was pilot tested with an individual who held a similar role to the intended participants but was not included in the final sample. These pilot interviews were used to assess the clarity, relevance, and sequencing of questions.

Based on the feedback, minor adjustments were made to improve the flow and ensure appropriateness for each audience (See interview guides attached as S1 Text, S2 Text, S3 Text). Interviews were aimed at generating perspectives regarding the context and mechanisms and outcomes of the intervention, including experiences of, (i) receiving, (ii) providing or (iii) developing fertility care at the centre.The interviews were scheduled consecutively, and data collection continued until all proposed sample size of participants was achieved. Although traditional data saturation was not a goal of this realist evaluation, data collection continued until no new Context-Mechanism-Outcome configurations emerged, and sufficient explanatory depth was achieved. Interviews were conducted in either Arabic or French based on participants preferences and lasted between 45–60 minutes. Researchers double checked that the information provided by the participants was as intended and whenever necessary, the researchers sought feedback from participants to clarify what they meant. Both female researchers were unknown to the participants prior to the study commencement. The interviews took place from the 5th of January 2023 to the 22nd of March 2023.To complement the qualitative data collection, data from the health information system was added to the data set to demonstrate clinical outcomes.

## Data analysis

The audio files of the interviews were transcribed by the researcher in full and a quality check of the transcription was performed. Inductive and deductive thematic analysis was conducted to identify explanatory middle range theories explaining

how and why the intervention works, based on the emerging themes related to the context, mechanisms and outcomes of the intervention. Two researchers coded independently the set of interview transcripts then compared their coding looking for differences and refining the codes. Any discrepancies were discussed and resolved collaboratively. Data were analysed manually using a thematic coding approach. Transcripts were read and segments of the transcripts were assigned to nodes which is a collection of similar ideas, related to the main domains of the CMO model, i.e., whether they related to the context, mechanisms or outcomes. These nodes were manually populated and then grouped into larger codes which were iteratively labelled to identify emerging subthemes that linked the outcomes (or impact)of the ART centre with either the context or specific mechanisms which were operating. Several interlinked subthemes were then iteratively modified and grouped to create overall themes under each domain of the CMO model. This process was repeated and discussed iteratively while remaining open to discovery, as is the practice in thematic data analysis [29]. The iterative process involved discussions between AB and GM which enabled the codes, subthemes, and overarching themes, to be organized and mapped onto the Context-Mechanism-Outcome (CMO) framework. The resulting CMO configurations were presented in table and diagrammatic format to illustrate key patterns across the different domains.As appropriate, the study utilised the WHO building blocks of health systems [18] to describe health system context and the development of each of the elements of the building blocks, namely service delivery, health workforce, health information system, essential medicines and equipment, financing, and leadership and governance. Data related to clinical outcomes was summarised descriptively.

### Data presentation

Quotations were used to support and illustrate the identified CMO configurations. Each quote is attributed to a participant category—couple, healthcare provider, or stakeholder—and assigned a unique, de-identified code for reference. Summary tables were used to present data related to clinical outcomes.

### Ethical approval

The study was approved by the Ethics Committee of Biomedical Research of the University Mohammed V of Rabat, Morocco, under reference: 96/22.

## Results

The study revealed information explaining how this intervention works, for whom, and under what circumstances, by shedding light on the context, mechanisms and outcomes of the ART Centre. Thematic findings related to CMO are described and summarized in Table 3.

Fig 1 shows a CMO Framework diagram that maps the themes and subthemes explicitly connecting context, mechanisms, and outcomes.

### Context

**Pronatalist culture which values childbearing and shuns childlessness.** Data showed several features of the circumstances under which the public ART centre was introduced, which were relevant to the operation of the program mechanisms. These contextual features included wider social, economic, technological, and demographic context nationally and were not necessarily to do with the location or setting of the centre itself. To start with, data indicated that the social expectation to bear children was high, which may have influenced the willingness of the population to interact or participate in the public ART centre.

*For society, children are the pillar of the family in our Moroccan context; any marriage must result in children…*" (Couple #1)

**Table 3. Themes and sub-themes relate to the CMO configuration of the intervention.**

| CMO Domain | Themes | Sub-themes |
|---|---|---|
| *Context* | Pronatalist culture which values childbearing and shuns childlessness | • Social expectation of childbearing<br>• Stigmatization of childlessness<br>• Psychological consequences (stress, anxiety, isolation) |
| | Varying awareness and prioritization of infertility | • Low prioritization of infertility<br>• Lack of political awareness and limited policy action<br>• Private sector dominance. |
| | High cost of treatment in the private sector | • High out-of-pocket costs in private sector<br>• Fundraising/ property sales to finance infertility treatment<br>• Motivation to seek affordable public services |
| *Mechanisms* | Practices of medical interactions andconsultations | • Trust-building and safety in consultations<br>• Psychological support and counselling<br>• Confidential, non-judgmental care<br>• Patient empowerment via self-care training<br>• Multidisciplinary teamwork<br>• Standard operating procedures and protocols<br>• Coordination with other HMSRO services and Ibn Sina Hospital departments |
| | Advance planning and strengthening of health systems | • Specialized training and capacity-building<br>• Acquisition of infrastructure, equipment, and medicines<br>• Integration of Health Information System (Gyneco2000)<br>• Bundled payment model<br>• Local Hospital (HMSRO) support for sustainability |
| | Enabling policy and legal environment | • Development of national ART law (Law 47–14, 2019)<br>• Advocacy for drug/consumable registration<br>• Advocacy for insurance coverage<br>• Strategic alignment with reproductive health policies<br>• Persistent gaps: lack of registry and licensing body |
| *Outcomes* | Access and affordability of ART | • Reduced cost relative toprivate sector<br>• Geographical barriers remain<br>• Need/ call for expansion across regions |
| | Positive patient experience and psychosocial impact | • Emotional wellbeing<br>• Quality of interpersonal care<br>• Improved trust in public health system |
| | Knowledge transfer and expertise | • ART training during residency<br>• Pilot for other public centres (Marrakech, Tangier, Oujda)<br>• Exchange visits and protocol sharing |

The expectations to have a child created immense pressure on couples, who regularly had to face questions related to having children and ended up being in social isolation. As a result, one participant explained that she would typically "*avoid visiting family and especially during happy events of marriage or baptism...to avoid questions from my family and acquaintances around the subject*" (Couple #3).

Data suggested that childbearing, including at an early age, was increasingly common and expected, and that delays in it get the couples especially women ostracized and isolated.

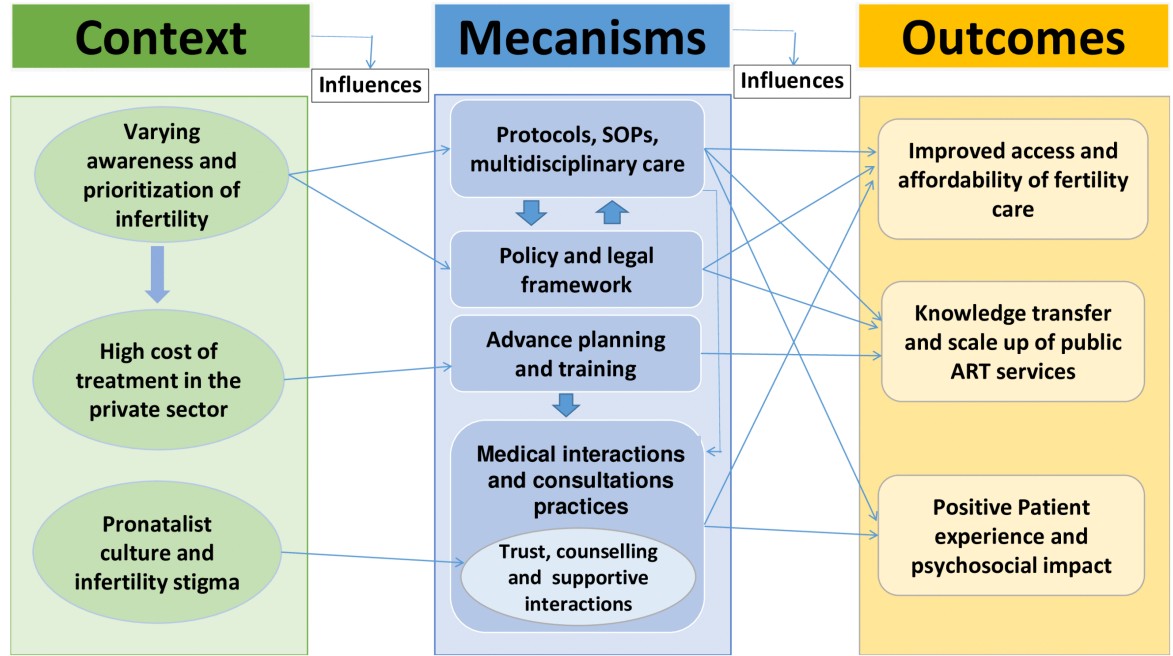

**Fig 1. CMO Framework diagram.**

*In our context, the age of marriage has increased, so it is a priority not to waste women's time before having a child, which is why we have prioritized this issue [of infertility]* (Stakeholder #1).

In addition, stigmatization of childlessness was rife, which created pressure for couples to seek solutions:

*I will never forget the pain I felt when I went to visit a relative who had just given birth... when I arrived her mother hid the baby so that I could not see it...as if I would bring bad luck…*(Couple #13).

This social pressure and stigmatization result in various psychological consequences including depression, anxiety, stress, low self-esteem, and anger. In a typical response, one participant stated that "*every time I think about it, I feel a lot of anxiety, stress and anger…*" (Couple #2). As a result, couples were generally eager to access treatment for infertility due to the prevailing social environment related to infertility.

**Contrasting awareness, visibility and prioritization of infertility.** Stakeholders suggested that the problem of infertility was common and based this assertion on recent surveys. Although the ubiquity of infertility was an important background circumstance that encouraged or enabled stakeholders to assemble and negotiate regarding how the problem could be solved, divergent views emerged in relation to the extent in which infertility was prioritized. On the one hand the ministry indicated that "*this [infertility] is a priority for the Ministry, and it has been included in the Ministry of Health's sectoral strategic plan.*" (Stakeholder #1). On the other hand, however, most policy makers indicated that infertility has not been accorded similar priority as to other health conditions.

*In my opinion and from my experience, infertility in Morocco has not been a national priority…we have worked in other priority health issues where state intervention has been decisive in their success.* (Stakeholder #5).

This viewpoint was supported by another stakeholder who asserted *"...the first obstacle is the lack of awareness by politicians".* (Stakeholder #3). Other stakeholders suggested that health authorities were more focused on birth control than fertility, and as a consequence, fertility care is much more developed in the private sector.It was asserted that it was only after 2013, that the Ministry of Health began to take an interest in this pathology and start to organize its management in the public sector.Thus, while infertility was visible in the private sector, it was not prioritized in the public sector.

*It is true that it's only since 2013 that the Ministry of Health started to develop this component of infertility... the management of infertility has been more developed in the private sector.*(Stakeholder #1).

**High cost of treatment in the private sector.**  An important contextual finding was the fact that the existing treatment facilities were mostly in the private sector where patients had "*spent millions between consultations, blood tests and medicines which were very expensive and non-reimbursable*" (Couple #10). It was typical to hear how patients*"had to repeatedly stop our follow-up with the attending physician because of the lack of money*" (Couple #11). As was the case with this couple, many patients had to fundraise from sale of property or loan from a family member in-order to access treatment from the private sector.

Not surprisingly, the prospect of accessing potentially cheaper treatment was an important underlying factor that influenced the decisions of couples to participate at the ART centre:

*We saw the inauguration of the centre on television, and we said why not, it's an opportunity to seize, especially since the price offered by the centre is much lower than that of the private sector... In the private sector, it's impossible for us to do IVF with their exorbitant prices which exceed our financial capacity by far.*(Couple #10).

## Mechanisms

**Practices of medical interactions and consultations.**  We found evidence showing that practices of medical interactions and consultations influence how the intervention is utilised and how it works. Our data suggest that the key to the successful implementation of our intervention is the concept of trust and safety. Couples' feelings of trust, and safety develop throughout the pre-treatment and treatment phases through constant accompaniment and psychological counselling by nurses:

*Listening and trust play a crucial role in the management of infertility; infertile couples are deeply traumatized and weakened psychologically and financially. At the centre, I feel that they consider that, and that influences the results.*(Couple #10).

This was also acknowledged by service providers:

*Infertile couples suffer enormously; a large part of their management is devoted to listening and psychological support.* (Healthcare Provider # 6).

Through constant repetition, clear communication and reinforcement of trust in subsequent visits, couples feel accepted, respected and valued and this was a highlight of couples' experiences:

*The verification of understanding, the fluidity of communication and constant psychological support, these elements marked my experience at the centre which "al hamdollih" was successful with the arrival of my little one*. (Couple #10).

Nurses enhanced psychological safety by safeguarding confidentiality while providing non-judgmental support. Conversely, 'unsafe' interactions would include practices that cause couples to experience embarrassment, rejection, and

stigmatization or blame for having infertility, which can ultimately impact on couples' feelings of trust and safety and thus, on their willingness to attend, participate in the public ART centre.To ensure that couples were further empowered, nurses trained the women to increase their self-efficacy for self-care in infertility. Self-care consists of helping women learn how to self-inject during their treatment regardless of their level of education. Participants indicated that this approach had a major role in empowering women during their ART treatments to adopt new behaviours, manage stress by going to relaxation and sports centres, and learn from other people's experiences, and making new connections during group therapy sessions:

*We set up the nursing consultation and self-care sessions which allow couples to be at the centre of the care and to have an active role in their own treatment*. (Healthcare provider # P5).

Participants suggested that standard operating procedures played a key role in streamlining the ways in which patients interacted with services. Long before the beginning of the clinical activities, the centre's team has drawn up protocols and procedures that concern clinical and biological acts related to the fields of nursing, medical and laboratory practices and compiled them in a manual:

*We started by developing norms and standards of practice here at the centre. The team generated protocols first before starting*. (Stakeholder #3).

The objective was to unify the practices in order to allow a homogeneous and fluid activity between the different members of the team; and to offer the couples a quality service delivery that involved multidisciplinary approach according to international standards:

*The management of infertility calls upon a multidisciplinary team and a specific technical platform... which implies standards, work sheets, strong coordination and the availability of quality drugs and medical devices.*(Healthcare provider #4).

Standard operating procedures between the ART centre and the other medical, technical and administrative services of the HMSRO were also developed, which facilitated the interactions and patient flow with the pharmacy department, the biomedical and technical department and the billing department. Worksheets were developed allowing optimal patient monitoring, better traceability and effective participation of couples in therapeutic protocols. Being part of CHUIS, coordination between the ART centre team and other CHUIS departments was also established especially with the Central Laboratory and the Samples Transport department. The integration of ART services at an already established public hospital was viewed positively by patients:

*My wife and I were very relieved. I believe in the good quality of the services in public hospitals which are non-profit, the medical decision is well deliberated and carefully thought out.*(Couple #1).

Expectations of good quality care was also expressed among the patients who were on the waiting list, who were "*full of a lot of hope*" and who reiterated that they "*expect a good quality of care and a good result* "(Couple #1).

Although the centre had developed structure and operating procedures, it was clear that a more widely national guideline for the management of infertility in Moroccan couples developed by the Ministry of Health was still needed, to complement the centre's operating procedures. The guideline will enable Moroccan health professionals, at all levels of the healthcare system, to provide standardized services before having recourse to ART. This guideline will complement a handbook on the standards of an ART centre whose purpose is to inform health professionals about the necessary standards for a fully operational ART centre.

*We have already drafted a national manual of public ART centres, but we have not yet published it since we are waiting for the relevant application texts to appear in the Law.* (Stakeholder #1).

**Advance planning and strengthening of health systems.** Through our interviews with stakeholders, it became clear that in order for the public ART centre to function effectively, couples had to feel confident that they would get the services that they need, and which required the infrastructure to be ready even before the public launch of the centre. Data showed that having the ART situated at the hospital had an impact on the couple's reasoning and their choices to participate at the centre. Aspects that required advanced planning included training of healthcare providers, acquiring the necessary equipment, as well as consumables such as drugs and medical devices.

*This centre began its work before the ceremonial inauguration event and made sure that people were trained in advance, that materials were ordered, that equipment was installed, that the structure was built, that the technical and human resources were prepared in advance and even tested over a period of 2–3 years before the ceremonial inauguration event.*(Stakeholder #1).

Evidence showed that deliberate strengthening of health systems, ranging from health workforce, essential medicines and equipment, financing, and leadership and governance contributed to the working of the centre. Enhancing the availability and skills of health care providers was an important underlying process that explained how the intervention was taken up:

*Specialized training in infertility does not exist in Morocco...we have benefited from this training which has ensured sustainability and successful implementation of the ART centre. It has been imperative to establish this training of doctors, biologists and nursing staff.*(Healthcare Provider #6).

Data showed that having infrastructure, medical supplies and equipment required significant investment but was essential in ensuring that the centre functioned appropriately, yet required advanced planning to ensure that these were all in place before construction of the centre was completed.

*After our return from training in Belgium, we set up everything from scratch, there were no standards for architecture, working procedures, management procedures, or technical specifications for the necessary materials and equipment …. We did all of this ourselves.* (Healthcare Provider #3).

The ART centre installed a health information system to enhance fluid management of couples' medical files and currently *"uses* [Gyneco 2000] *software, which was offered as part of the project by the ART centre of Erasme Hospital".*(Health care provider #3).

Stakeholders indicated that having all the required supplies in one place meant that the couples did not have to visit different venues for supplies, medications or tests. This simplified how couples accessed the treatments, which was perceived as having improved compliance and cost.

Findings suggested that the centre's payment system, which operates on a bundled payment model, covering the entire treatment cycle, could simplify billing and encourages treatment commencement or completion, as for couples would know how much the treatment would cost at the commencement of treatment:

*We made a model... a package including all the requirements in the service... the centre played a role in freeing the couple from the constraint and the difficulty of gathering everything they needed to do ART* (Stakeholder #3).

In addition, data showed that the local support and leadership with the main hospital where the centre was integrated was essential in reducing the cost of treatment and creating re-assurance for patients. Although the initial financial support was acquired from the Belgian Academy of Research and Higher Education, support was also provided from CHUIS, which ensured sustainability:

*The bilateral cooperation helped us with half of the financing…. the other half was covered by the hospital's own budget from the revenue generated after the financial balance between expenditure and revenue.* (Healthcare Provider #5).

Data also suggested that leadership was essential in galvanizing support and also creating governance of the sector. The HMSRO took the initiative to include in its Hospital Establishment Plan 2008–2012 the creation of the first public ART centre becoming the starting point of the development of fertility care services in the public sector.

*Prior to the establishment of the ART centre, the only player in the management of infertility and ART was the private sector...but it did not play the role of regulator;whereas when ART was introduced in the public sector, ART became more regulated, and we played a role in drafting the text of the law.* (Stakeholder #3).

As may be deduced from above, the leadership and governance model which engaged multiple stakeholders, with oversight from the Ministry of Health, enabled the centre to contribute to national policy development.

**Enabling policy and legal environment.** Stakeholders highlighted the importance of having the necessary legal, strategic and financial mechanisms in place, as well as a sustained commitment to develop them, as these processes require time and resources. Data showed that these policies, while often invisible, form important mechanisms which provide the legitimacy for service provision and financing. Our analysis has unpacked this mechanism further, highlighting the importance of having approvals for marketing authorizations, registrations and financing of ART supplies. Although the initial clinical and laboratory supplies, such as culture media and medical devices, were initially purchased using funding from the Belgian project supporting the centre, sustaining these supplies required approval and financing from the HMSRO.However, direct purchasing by HMSRO was initially impossible and *"the project had to suspend the laboratory activities while waiting for the registration of these products…"* (Healthcare Provider #3).This led to targeted advocacy with the Ministry of Health to accelerate the regulatory processes:

*We made a plea to the Ministry of Health for the acquisition of marketing authorizations, we fought to speed up the registration of culture media and medical devices which were not even classified by the Drugs and Pharmacy Department…for suppliers we constituted a limited market not worth the effort… the regulation and registration of the culture media and medical devices on which we worked has made it possible to motivate them"* (Stakeholder #3).

Financial sustainability was supported through advocacy efforts with the National Health Insurance Agency, in collaboration with the Centre's team, to secure reimbursement for some infertility drugs. Efforts are ongoing to include all ART treatments:

*We made a plea to the National Health Insurance Agency to include infertility drugs in the list of drugs reimbursable by the Compulsory Health Insurance …and currently there is a list in which infertility has been included. We are advocating, and work is being done by the National Health Insurance Agency to ensure the global management of infertility.* (Stakeholder #1).

Strategic coordination extended to national reproductive health policies, and policy makers incorporated infertility into the national reproductive health strategies 2011/2020 and 2021/2030. The Ministry of Health's 2025 plan mandated the establishment of ART centres in all University Hospitals, and data suggests that HMSRO had a role in this expansion:

*The HMSRO played a ground-breaking role in introducing this into the public sector and subsequently there is the University Hospital of Marrakech followed by the 2025 plan that the Minister of Health introduced because each University Hospital must have an ART centre. We were pioneers in this field,and we tried to transmit it elsewhere.* (Stakeholder #3).

Stakeholders emphasised that until recently, there was no legislation regulating ART practices. *"The initiative to develop a law had been underway since 2013…but it took a long time from 2013 to 2019, and it wasn't until 04/2019 that law 47/14 was passed"* (Stakeholder #1).The enactment of Law 47–14 in 2019 was a major milestone, which formally regulates ART practice nationally. Stakeholders highlighted that advocacy efforts were essential to accelerate this process and to overcome regulatory hurdles.

*Since April 2019, law 47/14 was created to regulate the practice of ART, its implementing texts are being developed.* (Stakeholder #3)

However, stakeholders acknowledged that despite continued efforts some areas were still lagging behind nationally, with important gaps remaining such as the absence of a national ART registry and licensing body, which are foreseen but not yet implemented under Law 47–14:

*At this moment there is no national registry or licensing body, but it is foreseen in Law 47-14.* (Stakeholder #4).

Together, these policy, legal, financial, and advocacy measures constitute the enabling environment that actively supports the establishment and ongoing operation of public ART services in Morocco.

## Outcomes

**Access and affordability of ART.** The cost of ART cycles at the centre are detailed in the S4 Table which reflects the full amount paid directly by the couples, as there is currently no insurance coverage or state subsidy for ART in Morocco. Data suggested that the intervention was perceived as having an impact on the accessibility and affordability of ART:

*It's an improvement in the accessibility of the population and the practice of ART, particularly for the vulnerable population. The management costs much less especially when compared with private centres,…which means that this centre has contributed to improving accessibility for couples who don't have the [financial] means.*(Stakeholder #1).

Another stakeholder added that:

*The pilot ART centre has improved the access to services for couples who do not have the means and reduced their psychological suffering. It provides a holistic care of good quality with a reduced cost of 50% compared to the private sector and in the same structure, offering more comfort to couples.*(Healthcare Provider #5).

All the couples have already consulted in private clinics before coming to this public centre. In response to what are the existing advantages between a public ART centre compared to a private centre, they declared it is above all the difference in cost which is cheaper in the public centre. All the couples declared that they would recommend the centre to their family and acquaintances.

*The cost is much cheaper compared to the private sector, which is very high, because a private doctor asked me for a very high price. I felt that there is a lot of hope opening up.* (Couple #4).

A stakeholder added that:

*It brought a solution to infertility; it is about accessibility especially financially; the cost is cheaper.* (Healthcare Provider #1).

However, other factors such as travel, and lodging expenses still constitute major obstacles for infertile couples showing the importance of having other ART centres in the other regions of the country.

*I come from very far and I spend the whole night on the way to arrive on time at the centre and these are repeated follow-up visits...I have nowhere to spend the night in Rabat and I do not have the means to pay for short term accommodation… Unfortunately, there is no such centre in our region.* (Couple #7).

This view was also supported by a couple who stated that:

*We must provide guest houses near the centre to improve geographical accessibility or provide hotels at affordable costs and multiply other ART centres.* (Couple #1).

Stakeholders agreed that while the ART Centre had succeeded in making ART more affordable, they highlighted that cost was still an obstacle in the management of infertility more widely in Morocco, partly because the Compulsory Health Insurance does not include the treatment of infertility.

*Infertility was not considered when setting up compulsory health insurance in Morocco... this exposes couples suffering from this health problem to catastrophic expenses and great financial vulnerability.* (Stakeholder #5).

This sentiment was shared by couples who suggested that to improve the services, it is necessary to enhance "m*edical coverage for the management of the disease, and the reimbursement of drugs*…" (Couple #4). In addition, most couples suggested that the government should "…*replicate similar centres to improve geographic accessibility in other regions to limit the movement of people residing in distant cities*..." (Couple #3). Stakeholders also supported this viewpoint. To illustrate, one opined that *"...despite the efforts made to promote infertility......couples from remote areas must absolutely travel to the city where the service is available..."* (Stakeholder #3). Overall, however, most of the responders agreed that the public ART centre has succeeded and that the Moroccan experience of creating it a should be transferred to other countries so that "*all couples across all countries should benefit from the services of such public ART centres*…" (Couple #1).

**Positive patient experience and psychosocial impact.** Patient testimonials illustrated the positive emotional outcomes associated with the public ART centre. One couple shared that:

*Before arriving at the centre, we consulted several doctors and private clinics. Unfortunately, the interest in the private sector was financial, and we did not feel supported. On the other hand, at the centre, we felt well taken care of, with special attention to each couple and close follow-up. We felt important and that our problem truly mattered to the team. Moreover, the cost of care is affordable, and the centre's doctor is very empathetic and attentive. I recommend the centre to every woman suffering from this problem because here she will receive the correct information and appropriate, quality care.* (Couple #10).

This testimonial emphasizes the positive impact of the program on patients' emotional wellbeing, highlighting how compassionate and attentive care at a public centre can foster a sense of being valued and supported. Several other

participants also identified how empathetic follow-up and affordable services helped rebuild the couple's confidence in the public health system. By providing respectful, personalized care and transparent information, the program plays a critical role in reducing the stigma associated with infertility and empowering couples throughout their treatment journey as was expressed by a participant "*The treatment is not easy, but I learned how to do my own injection and understand the stages of the treatment at my own pace*". (Couple #8).

**Knowledge transfer and expertise.** The creation of the ART centre in 2013 was the first step into providing a specialized training in ART in a university hospital setting.

*It is the ART centre of the Maternity of Orangers Hospital that introduced ART in the basic curriculum of residents.* (Stakeholder #3).

In addition, and due to the importance of having ART centres all across the country. The ART centre represented a pilot project for other centres.

*For the other public centres,there was knowledge transfer. Centres in Marrakech, Tangier, Oujda have benefited in various ways from this centre.* (Stakeholder #3).

Similar view also emerged from healthcare providers who participated in learning exchanges:

*Our experience had helped the newer public ART centres whose staff came to visit the centre and equipped themselves with our guides and protocols... we supported their teams in the implementation and start-up of their new centres*. (Healthcare provider #6).

### Clinical outcomes

Six hundred and ninety cycles were performed including IVF (34.2%), FET (17.5%), IUI (4.2%), and ovulation induction (44.2%). Clinical pregnancy rate was 23.8%for fresh IVF cycles, and 21%for FET cycles. Outcomes from IVF and FET cycles are shown in Table 4. No severe OHSS or other complications were reported.

All treatment dropouts took place after initial consultations and diagnostic investigations, but before the start of any treatment cycle. No dropouts occurred during ovarian stimulation, after oocyte retrieval, or following embryo transfer.

There were very few indications for IUI. A total of 29 cycles have been carried out at the centre, most of which were in during the first few years of the implementation of the centre; since then there has been few indications for this technique. From the 29 IUI cycles, two cases were cancelled (6.9%). One pregnancy was achieved, resulting in a live birth.

Among the 305 ovulation inductions performed, 23.9% were cancelled primarily due to the absence of an ovarian response, or rarely the risk of a multiple pregnancy. The clinical pregnancy rate was 18.1%. Miscarriage rate was 28.6%, with one ectopic pregnancy. Live birth rate per initiated cycle (without cancelled cycles) was 12.5%. Detailed IVF, FET, IUI, and ovulation induction cycles' characteristics are available in S5 Table, S6 Table, S7 Table.

### Discussion

This study documents the processes and outcomes of setting up an ART centre in the public sector in Morocco, where, since the 1980s, the only existing ART centres had been private clinics. The aim of our study was specifically to investigate *how* the components of the program result into the intended outcomes of enhancing access to fertility care. Results showed that the creation of the first public ART centre in 2013 in the HMSRO in Rabat represented a key milestone enhancing fertility care provision in the public sector. Several insights emerge from our study's CMO analysis, which warrant discussion.

Table 4. Summary outcomes of IVF and FET cycles.

| Outcome | In vitro Fertilization (IVF) | Frozen Embryo Transfer (FET) |
|---|---|---|
| Number of cycles | 236 | 121 |
| Cycle Cancellation rate | 20/236 (8.5%)* | 16/121 (13.2%) |
| Embryo Transfer rate | 151/236 (64%) | 105/121 (86.8%) |
| Cryopreservation rate | 140/236 (59.3%) | – |
| Clinical Pregnancy rate/ transfer | 36/151 (23.8%) | 22/105 (21%) |
| Live birth rate/ transfer | 25/151 (16.6%) | 15/105 (14.3%) |
| Multiple Pregnancy rate | 5/36 (13.9%) | 4/22 (18.2%) |
| Miscarriage rate | 10/36 (27.8%) | 6/22 (27.3%) |

*Cycle Cancellation rate before Oocyte pick up.

First, in terms of the context, our study shows that on the one hand, the confluence of social expectation to have a child and stigmatization of childlessness can enhance engagement with ART services, while on the other hand, the high cost of IVF and low prioritization of infertility hinders engagement with ART services. The context in Morocco is broadly similar to many LMICs (albeit with few cultural, religious and legal variations), and it is therefore not surprising that political (for example lack of prioritization of understanding that it is a serious problem [12,14,15,30–32]), health system (such as distances [33]), and social-cultural and religious factors (such as lack of awareness [34,35], and social values [36]) have been shown to influence the extent to which populations are motivated to engage with and seek help from infertility services, while at the same time, contributing to stigmatization, secrecy [37] and economics hardships [38] among couples with infertility who seek help.

Second, within this context, our study shows that several mechanisms are critical to the success of Public ART services. These include leadership and the way that fertility care is planned, organized and provided. Leadership and involvement of the health authorities was critical in ensuring that the health systems are strengthened to provide these services. Leadership was essential in galvanizing support and creating governance of the sector enabling policy and legal environment. Studies from other countries, such as Brazil, have previously demonstrated how a lack of political decision to implement ART services, and the lack of allocation of public finances devoted to infertility care is a chief barrier to fertility care [39]. A recent systematic review also identified low political commitment as a key barrier to inclusion of fertility care in reproductive health policies in the African continent [31]. At the same time, our results show that while the first obstacle to the development of public ART services is the lack of awareness by politicians that infertility is an important issue that requires policy prioritization and funding, once that is addressed, there must be efforts to strengthen all aspects of the health systems to enable service provision. Our study shows that investments in health systems infrastructure, supply chain, and training of health care providers are key mechanisms for successful implementation of ART in the public sector. This is consistent with a review which found that successful implementation of ART services "*requires a specialized, organized medical and paramedical staff and a minimum of infrastructure within the health system*", including "*supply of materials and improvement in existing services*" [40].

The enactment of the law 47–14 on ART, constituted an important mechanism supporting the ART centre. The promulgation of ART law provided a framework mechanism within which health systems could be strengthened and ART safely provided. Once infertility was included in the national health strategy, our study shows that advanced planning and mobilization of resources was an important next step and mechanism. Although participants in our study considered the cost of treatment in the ART centre to be cheaper compared to private clinics, costs at our centre may still be costly for some couples because health insurance does not cover full costs of treatment. The absence of health insurance coverage, whether private or state-funded, was highlighted by responders as a significant factor in the success of the intervention.

Advanced planning and strengthening of health systems was highlighted by stakeholders as a crucial mechanism in the successful creation of the centre. For instance, advanced planning was required to train health professionals, because, as in many countries, courses in reproductive medicine and biology were not previously included in training curricula in university teaching hospitals and medical schools in Morocco [24]. This lack of skills necessary to practice ART by health professionals is present in many LMICs, which forces clinicians to travel to other countries to "*acquire the knowledge; refer to European or American guidelines; and organize [training] collaborations with European centres*" [41]. Additionally, the quality of medical interactions and consultations emerged as a critical mechanism that increases the acceptability of ART from public facilities. The quality and the humanization of medical interactions can facilitate patients' adherence to treatment, and sense of trust.

In addition, the ART centre has implemented quality assurance mechanisms such as Standard operating procedures (SOPs), clinical and laboratory protocols that enable safe and effective service delivery, and a health information system that facilitates secure and confidential management of patient data which contribute to some of the mechanisms supporting the centre's outcomes. The centre also implemented two studies to establish and validate internal quality control for sperm concentration measurement [42] and to assess ART clinical outcomes to inform service improvements [43]. These efforts reflect a commitment to continuous quality improvement.

In terms of outcomes, data suggested that the intervention was perceived as having a positive impact on the accessibility and perceived affordability of ART which resulted in 2,495 couples accessing treatment at the centre. However, most respondents acknowledged that the next step must be the creation of at least one public ART centre per region in a university hospital in other parts of the country to further increase access, which will further decentralize these services. Our study suggests that geographical inaccessibility is an additional barrier influencing the decision of couples to seek or not the services of the ART Centre. For instance, our centre served fewer participants coming from distant regions compared to those near the centre, possibly due to transport, travel, accommodation and catering costs.

While our ART Centre has been successful in increasing access and perceived affordability and trust in public ART services, some clinical outcomes need improvement. IVF and FET cycles resulted in a clinical pregnancy rate of 23.8% and 21% respectively, and a live birth rate of 16.6% and 14.3% respectively. Notably no dropouts occurred during ovarian stimulation, after oocyte retrieval, or following embryo transfer. This continuity was likely facilitated by the patient engagement and interactions, as well as payment model, in which each ART treatment cycle is covered as a single bundled package, encouraging patients to complete the full course of treatment and follow-up (S4 Table). Nevertheless, age-specific success rates and dropout information remain partial. Additionally, time-trend data are limited. These are critical metrics for full assessment of the program effectiveness and relevance and these gaps need to be prioritized in future data collection.

Besides outcomes related to patients, institutional, policy and legal environment, broader outcomes related to equitably access to services can result from the establishment of public ART services.Prior to the public ART centre, infertility care in Morocco was wholly provided by private sector clinics, which is often associated with high out-of-pocket costs and limited regulation, especially in Morocco and other LMICs [38,44]. The introduction of a public ART centre represented a significant disruption to private sector monopoly by offering an alternative, more affordable and regulated source of care, which can drive competition and gradual reform.Furthermore, addressing infertility in public hospitals, and within government policy can shift public perception of infertility,assist to destigmatize it, and mitigate financial and gender inequities when services for both men and women and couples are promoted [13]. The couple-centered approach implemented in the Public ART Centre was meant to ensure couples understand that infertility is a shared reproductive health issue, and encourage joint engagement in care, given that women are generally blamed for infertility in our context. While we do not have clear evidence that our centre achieved wider social-structural outcomes, studies have shown that changing counselling strategies and ensuring that providers of interventions include both men and women can have an impact on community perception, masculine norms and infertility stigma [45].

                                                          

## Implications for policy, services replication, and future research

Our study highlights the need to expand public ART provision in Morocco to enhance geographic accessibility, financial affordability, and service quality. Currently, there are 3 public centres across the country. Considering that the estimated need for ART (including IUI and IVF) is 1500 cycles per million people per year [46], it is possible that Morocco will need more cycles, given that the proportion of the Moroccan population aged 15–59 is predicted to reach 63.1% in 2030 [47]. While ensuring at least one ART centre per region is an important long-term goal, resource and capacity constraints may not always make standalone centres feasible. Alternative models include embedding ART units within existing multispecialty university hospitals, developing regional ART hubs linked to primary health care centres through tiered referral protocols, and using teleconsultation platforms to extend specialist support to remote areas. Such models allow progressive scaling of services while optimizing existing infrastructure and human resources, thereby ensuring equitable access without overstretching the health system.

Additionally, although the national ART law and strategy are important mechanisms that are facilitating public provision of ART, the establishment of a National Registry of ART is needed in order to ensure that clinical data and health information related to ART services can be used to inform services and policy decisions in a uniform, transparent and harmonized way [48]. Although, to date there is no national registry of ART, there is a strong awareness of its importance among stakeholders. Experiences from other countries such as Belgium, and South Africa [49] show that registries are vital for transparency, evidence-based policy, and clinical improvement.

Moreover, several complementary actions are needed to strengthen public ART services. These include reducing out-of-pocket costs through insurance coverage, investing in sustainable training and certification, defining clear fertility care pathways, and promoting public–private partnerships to optimize resources. Table 5 summarizes the key policy recommendations derived from our findings.

Beyond the Moroccan national context, our findings imply that replicating public ART models is possible across other LMICs. Although models of public provision of infertility care are rarely described in LMICs, a publicly funded reduced-cost infertility service was initiated within a maternal health department of a hospital which provides healthcare for military, police, and other security personnel in Rwanda [50]. It demonstrated how lower-cost ART services can be effectively integrated into public hospitals, showing that affordable care is feasible in resource-constrained settings [50].In south Africa a documentation of ongoing services at a university hospital showed the need to expand access in public sector [51].In The Gambia, a participatory workshop co-designed national priorities for infertility services, underscoring the importance of stakeholder engagement in service planning [52]. These examples support the feasibility of replicating public ART models across LMICs. In our own and Rwandan examples, developing partnerships with other institutions, and ongoing collaboration with national and international partners (for example in Europe or the USA) facilitated benchmarking against global standards, and supported capacity building.There are other examples of countries with developed ART provision and financing including the Netherlands and France, among others [53]which can also be part of international collaborations.

In terms of research, our study applies realist evaluation and identifies important components of the classical CMO configuration defined by Pawson and Tilley. As recently pointed out by De Weger et al. [28], several studies have added explanatory factors to the CMO configuration, for example the intervention–context–actor–mechanism–outcome (ICAMO) or the, strategy–context–mechanism–outcome (SCMO) configurations. In our application, actors and strategies were included in describing the context or mechanisms. Future studies related to ART could further explore the importance of these explanatory factors.

## Study limitations

The results of this study are dependent on the Moroccan context, which may affect the generalizability of findings to other settings. The outcomes of the intervention occurred in Morocco, with its set of social, cultural, economic, and political circumstances, and while there are similarities with other low- and middle-income settings, some differences may exist.

PLOS Global Public Health

**Table 5. Policy recommendations for strengthening public ART services in Morocco.**

| Recommendation | Rationale | Proposed Approach |
| --- | --- | --- |
| **Geographic expansion** | Only 3 public ART centres exist; Uneven access across regions. | • Establish regional ART hubs within multispecialty university hospitals<br>• Primary health care centres and district hospitals should act as "spokes" providing infertility screening, initial evaluation, basic management (ovulation induction, IUI), counselling, and follow-up.<br>• Complex cases (for instance requiring IVF/ICSI) should be referred to the regional ART hub.<br>• After ART treatment, patients can be referred back to local centres for pregnancy monitoring and general reproductive care.<br>• Teleconsultation between the hub and spoke centres allows specialists to guide local providers |
| **Insurance coverage** | High out-of-pocket costs limit equitable access to ART. | • Include ART services within Compulsory Health Insurance schemes.<br>• Ensure reimbursement mechanisms are clearly defined. |
| **National ART registry** | Lack of standardized data limits monitoring, transparency, and policy planning. | • Establish a national ART registry to ensure (i) uniform data collection, (ii) long term evaluation, and accountability, and (iii) equitable resource allocation and optimized care. |
| **Training, certification, and care pathways** | Shortages of trained ART professionals and fragmented infertility care pathways weaken service quality. | • Invest in specialized training and certification programs to support sustainable human resources.<br>• Define clear infertility care pathways integrated within the health system. |
| **Public–private partnership** | Public sector resources alone may be insufficient to meet demand. | • Promote collaboration with private providers by (i) facilitating joint training programs; (ii) sharing expertise and resources while ensuring affordability. |

Nevertheless, our study provides useful learnings that may inform the implementation of a public ART Centre in other settings.

## Conclusion

The creation of the first public ART centre represented a milestone in the management of infertility in the Moroccan health system with a complete and holistic care model. The centre has played a crucial role in enhancing accessibility, achieving comparable clinical outcomes, and contributing to the transfer of this technology to other settings nationally. However, despite many achievements in improving the management of infertility in Morocco, many challenges are still ahead. Nevertheless, learnings from this centre provides useful information that can inform the creation of public ART centres in other LMICs to ensure universal access to fertility care.

## Supporting information

**S1 Fig. Couples' care pathway.**

(PDF)

**S1 Text. Interview Guide for Men and Women with infertility.**
(PDF)

**S2 Text. Interview Guide for HealthCare Providers.**
(PDF)

**S3 Text. Interview Guide for Policy Makers.**
(PDF)

**S1 Table. COREQ Checklist.**
(PDF)

**S2 Table. Characteristics of the centre's population.**
(PDF)

**S3 Table. Timeline of the implementation of the ART centre.**
(PDF)

**S4 Table. Cost of ART cycles at the centre.**
(PDF)

**S5 Table. Characteristics of Fresh IVF cycles.**
(PDF)

**S6 Table. Characteristics of FET cycles.**
(PDF)

**S7 Table. Characteristics of IUI and Ovulation induction cycles.**
(PDF)

**S1 Data. Quantitative data.**
(XLSX)

**S2 Data. Transcriptions in English.**
(ZIP)

## Acknowledgments

The authors are grateful to all participants for their participation in the study.

## Author contributions

**Conceptualization:** Amal Benbella, Gitau Mburu, Karima Gholbzouri, James Kiarie, Rachid Bezad.

**Data curation:** Amal Benbella, Asmaa Zaidouni, Sanae Elomrani.

**Formal analysis:** Amal Benbella.

**Investigation:** Amal Benbella, Asmaa Zaidouni, Sanae Elomrani.

**Methodology:** Amal Benbella.

**Project administration:** Gitau Mburu, Rachid Bezad.

**Software:** Gitau Mburu.

**Supervision:** Gitau Mburu.

**Validation:** Asmaa Zaidouni, Sanae Elomrani, Gitau Mburu, Karima Gholbzouri, James Kiarie, Rachid Bezad.

**Writing – original draft:** Amal Benbella, Gitau Mburu.

**Writing – review & editing:** Amal Benbella, Asmaa Zaidouni, Sanae Elomrani, Gitau Mburu, Houyam Hardizi, Abdelhakim Yahyane, Hafid Hachri, Karima Gholbzouri, James Kiarie, Rachid Bezad.

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
