## [Decision Letter · Decision Letter 0]

14 Jul 2025

PGPH-D-25-01503

A realist evaluation of the development, implementation and outcomes of the first public ART centre in Morocco

Dear Dr. Benbella,

Thank you for submitting your manuscript to PLOS Global Public Health. After careful consideration, we feel that it has merit but does not fully meet PLOS Global Public Health’s publication criteria as it currently stands. Therefore, we invite you to submit a revised version of the manuscript that addresses the points raised during the review process.

We look forward to receiving your revised manuscript.

Kind regards,

Rahul Gajbhiye, MBBS PhD

Academic Editor

Journal Requirements:

1. In the online submission form, you indicated that [The datasets used and/or analyzed during the current study are available from the corresponding author on reasonable request.].

a. In a public repository,

b. Within the manuscript itself, or

c. Uploaded as supplementary information.

2. Please ensure that you provide a single, cohesive .tex source file for your LaTeX revision. You may upload this file as the item type 'LaTeX Source File.' As stated in the PLOS template, your references should be included in your .tex file (not submitted separately as .bib or .bbl). Please also ensure that you are making any formatting changes to both your .tex file and the PDF of your manuscript. If you have any questions, please contact Latex@plos.org. You can find our LaTeX guidelines here:

https://journals.plos.org/globalpublichealth/s/latex

3. “Ethics Approval AF 96-22.pdf and English translation of the ethics approval.pdf” are currently uploaded as an 'Other' file type, which is not viewable by reviewers. Please ensure that all files meant for review are uploaded as 'Supporting Information' and include a legend in the manuscript.

Additional Editor Comments (if provided):

Reviewers' comments:

Reviewer's Responses to Questions

**Comments to the Author**

1. Does this manuscript meet PLOS Global Public Health’s publication criteria? Is the manuscript technically sound, and do the data support the conclusions? The manuscript must describe methodologically and ethically rigorous research with conclusions that are appropriately drawn based on the data presented.? Is the manuscript technically sound, and do the data support the conclusions? The manuscript must describe methodologically and ethically rigorous research with conclusions that are appropriately drawn based on the data presented.

Reviewer #1: Yes

Reviewer #2: No

2. Has the statistical analysis been performed appropriately and rigorously?

Reviewer #1: Yes

Reviewer #2: No

3. Have the authors made all data underlying the findings in their manuscript fully available (please refer to the Data Availability Statement at the start of the manuscript PDF file)?

The PLOS Data policy requires authors to make all data underlying the findings described in their manuscript fully available without restriction, with rare exception. The data should be provided as part of the manuscript or its supporting information, or deposited to a public repository. For example, in addition to summary statistics, the data points behind means, medians and variance measures should be available. If there are restrictions on publicly sharing data—e.g. participant privacy or use of data from a third party—those must be specified.requires authors to make all data underlying the findings described in their manuscript fully available without restriction, with rare exception. The data should be provided as part of the manuscript or its supporting information, or deposited to a public repository. For example, in addition to summary statistics, the data points behind means, medians and variance measures should be available. If there are restrictions on publicly sharing data—e.g. participant privacy or use of data from a third party—those must be specified.

Reviewer #1: Yes

Reviewer #2: No

4. Is the manuscript presented in an intelligible fashion and written in standard English?

Reviewer #1: Yes

Reviewer #2: Yes

Reviewer #1: 1. Congratulations to the authors for their hard work. It is good work done of establishing the ART center in Morocco and a good role model for other LMICs.

2. The qualitative data is described in the manuscript however the details of quantitative data w.r.t services availed, success rates is lacking. It is mentioned in the beginning that quantitative data will be described.

3. Please check with the word limit of the journal for qualitative manuscript. Description can be made concise

4. In discussion please mention about models of care for infertility in LMICs, if published literature is available

5. remove typical from line 126 and 128

6. mention the details of ART services provided at the center

7. 129 line onwards As of the time of study used very frequently .

Comment: Minor Revision. Such Models should be published

Reviewer #2: 1. Does this manuscript meet PLOS Global Public Health’s publication criteria?

Response: No

My feedback:

The manuscript number Manuscript Number PGPH-D-25-01503 presents a promising model of public sector ART service delivery in Morocco and reflects a rare example of health system innovation in a traditionally privatized domain. However, it lacks critical elements that would allow it to serve as a model for replication, such as disaggregated outcomes, internal/external evaluations, adaptation learnings from the Belgian model, and structured quality assurance indicators. Therefore, while methodologically it holds promise, it is not yet ready for publication in its current form.

2. Has the statistical analysis been performed appropriately and rigorously?

Response: No

My feedback:

No statistical or quantitative evaluation of program impact is presented. The paper mentions 2,495 beneficiaries (Line 52) but does not clarify whether this refers to individuals or couples, nor does it provide clinical success rates (pregnancy, live birth, etc.). Time-trend data and baseline burden are entirely absent.

3. Have the authors made all data underlying the findings in their manuscript fully available?

Response: Unclear

My feedback:

No links to raw data, transcripts, or internal monitoring systems have been provided. There is insufficient transparency in how the findings (especially Table 3) were generated.

4. Is the manuscript presented in an intelligible fashion and written in standard English?

Response: Yes, with revisions needed

My feedback:

While the language is generally clear, specific terminology such as “official inauguration” (Line 381) needs replacement with “optimally functional” to better reflect when the center became fully equipped to offer services professionally. This also implicitly conveys the time needed when LMICs lack a formal enabling policy. The structure also suffers from repetition in the CMO framework across sections.

While the manuscript refers to the realist evaluation model, the connection between context, mechanism, and outcomes is not consistently or clearly demonstrated. Authors should strengthen how these three elements interact across different domains in Table 3 and the results.

5. Review Comments to the Author:

Strengths:

The paper captures a policy breakthrough in making ART services publicly accessible in Morocco, a country with a pluralistic health system.

The timeline from concept (2008) to operationalization (2013) reflects sustained effort and partnership with Belgian clinical institutions (Lines 398–404).

The use of realist evaluation offers valuable theoretical framing.

Table 3 is well-placed under results (Line 231) and integrates institutional and societal enablers.

Areas for Improvement:

Ambiguity in Sample Size and Units

Line 52: Mentions 2,495 “patients,” and Line 477 states “more than 2,000 people.” In ART, the unit is typically “couples.” Clarification is needed. This has implications for calculating success rates per treatment cycle, assessing access equity, and establishing appropriate denominators for ART program performance.

Gender Dynamics

Line 306–307 notes that male infertility was frequently observed at the center, helping raise awareness on a previously socially invisible issue. This is a critical gender-sensitive insight that deserves greater visibility in the Results or Discussion section. The manuscript could elaborate on whether this influenced the design of counseling protocols, community messaging, or clinical engagement strategies. A policy reflection on the role of public centers in reshaping gendered narratives of infertility would add depth and align with WHO's gender equity lens.

Inadequate Outcome Reporting

Lines 52–53: Mention the number of beneficiaries but omit clinical and programmatic outcomes such as live birth rate, age-specific success rates, number of treatment cycles, patient follow-up, and dropouts. This is a major limitation for a paper claiming success of a public sector ART model. Without such outcomes, replication, scalability, and cost-effectiveness analysis remain speculative.

Missing Baseline Burden and Trends

No reference is made to the infertility burden in Morocco, either from Demographic Health Surveys, national estimates, or global data sources (such as WHO’s 2023 report on global infertility burden). Absence of such trend data (2008–2021) makes it difficult to assess the program’s reach or relevance. The lack of historical context and service coverage dilutes the paper’s utility for policy and planning.

Especially important is the absence of dropout rates, failed cycles, and follow-up completion. These are core indicators for ART program performance and their omission undermines claims of effectiveness.

Inconsistencies in Timeline of Operations

Line 381: The term “official inauguration” is from the article and may be retained, but the review suggests clarifying when the center became optimally functional—that is, professionally staffed, regulated, and delivering quality-assured services. This will avoid reader misperception that initial services were makeshift or informal and more accurately reflect the delay caused by the absence of enabling policies in LMIC contexts.

Replication and Adaptation Unclear

Lines 398–404: The manuscript notes the replication of Belgian standards, but without clarity on contextual adaptations. What clinical protocols, eligibility criteria, or service delivery components were localized? Were any services modified due to sociocultural or systemic differences? These questions are central to understanding how LMICs can replicate such a model with adaptation. The paper should offer examples of which parts of the Belgian model were directly adopted and what was modified to suit the Moroccan context.

WHO Health System Pillars

Lines 107–108: The authors refer to WHO’s six health system building blocks but do not systematically address them. Key gaps include:

Health Workforce: No mention of staff training, cadre mix, retention mechanisms, or capacity building.

Health Information Systems: The existence of an HMIS is mentioned, but no data on indicators collected, gender disaggregation, or system use is provided.

Leadership and Governance: Decision-making, accountability systems, or institutional coordination (between MoH, local governments, and partners) is not discussed.

Health Financing: Cost per ART cycle, proportion of out-of-pocket expenditure, insurance mechanisms, or state subsidies are not documented, despite being critical to assess equity. The manuscript should also clarify whether any cost transparency measures were in place, such as fee disclosure to patients or standardized billing. This would signal good governance and equity.

No Mention of Quality Improvement (QI) or External Evaluation

There is no reference to any internal audit, clinical review cycle, patient feedback mechanism, or third-party evaluation. In a field as sensitive as infertility and setting up something greenfield project- quality assurance is central to patient safety, ethical practice, and public confidence. Future revisions should specify whether QI processes (e.g., treatment success audits, stock-out surveillance, complication monitoring) were instituted.

Impact Beyond Access – Results Underreported

The manuscript restricts its results to availability and access, ignoring broader systemic and societal impact. Suggested additions include:

Program influence on policy dialogue and normalization of ART as a public health service

Regulatory effects on pharmaceutical and consumable pricing or availability

Testimonials showing how the program affected patients’ emotional wellbeing, confidence in public health systems, or reduced stigma. These dimensions are especially important in infertility care.

Policy Recommendations Too Generic

Line 576: Recommending “similar centers” does not account for resource constraints, regional needs, or institutional capacity. Consider suggesting embedded ART units within existing multispecialty hospitals with teleconsultation support, tiered referral protocols, or regional ART hubs linked to PHCs.

Professional Organizations & Multi-Stakeholder Voice

Line 84: Absence of public sector models is noted but not analyzed. There is no mention of how the initiative engaged (or did not engage) professional bodies like gynecology societies, community representatives, or pharmacists. The governance and advocacy ecosystem is therefore underexplored.

Terminology

“Enabling environment” in Table 3 remains vague. Authors should illustrate enabling conditions through actual programmatic measures—such as provider incentives, inclusion of infertility in national policy, mitigation of religious/cultural opposition, and streamlining of supply chains for ART consumables.

It would further help to list concrete operational mechanisms like subsidy mechanisms, task-shifting authorizations, or partnership MOUs with local governments that made this “enabling environment” functionally meaningful.

Private Sector Monopolies – Missed Opportunity

The paper misses the opportunity to reflect on how ART services in many LMICs are monopolized by private providers with high out-of-pocket costs and minimal regulation. A public sector entry not only enhances access but challenges traditional market structures. This critical policy insight should be emphasized.

Comparative Reference to France

While the manuscript notes Belgian collaboration, it could benefit from referencing France’s publicly funded ART system, given linguistic and policy parallels in North Africa. Comparative insights would elevate the manuscript’s relevance to other middle-income countries. France offers a publicly funded, equity-oriented ART program, and its health regulations and social norms are closer to the Maghreb context. Referencing its success could guide policymakers in similar sociocultural regions.

**Do you want your identity to be public for this peer review?** For information about this choice, including consent withdrawal, please see our Privacy Policy..

Reviewer #1: **Yes:** Dr Anushree Devashish PatilDr Anushree Devashish PatilDr Anushree Devashish PatilDr Anushree Devashish Patil

Reviewer #2: No

---

## [Decision Letter · Decision Letter 1]

19 Jan 2026

PGPH-D-25-01503R1

A realist evaluation of the development, implementation and outcomes of the first public ART centre in Morocco

Dear Dr. Benbella,

Thank you for submitting your manuscript to PLOS Global Public Health. After careful consideration, we feel that it has merit but does not fully meet PLOS Global Public Health’s publication criteria as it currently stands. Therefore, we invite you to submit a revised version of the manuscript that addresses the points raised during the review process.

We look forward to receiving your revised manuscript.

Kind regards,

Rahul Gajbhiye, MBBS PhD

Academic Editor

Journal Requirements:

Additional Editor Comments (if provided):

Reviewer 1 : Most of the comments have been included as possible. Word limit needs to be checked to make it concise. Few grammatical checks required.

Reviewers' comments:

Reviewer's Responses to Questions

**Comments to the Author**

Reviewer #1: All comments have been addressed

publication criteria? Is the manuscript technically sound, and do the data support the conclusions? The manuscript must describe methodologically and ethically rigorous research with conclusions that are appropriately drawn based on the data presented.? Is the manuscript technically sound, and do the data support the conclusions? The manuscript must describe methodologically and ethically rigorous research with conclusions that are appropriately drawn based on the data presented.

Reviewer #1: Yes

3. Has the statistical analysis been performed appropriately and rigorously?

Reviewer #1: Yes

4. Have the authors made all data underlying the findings in their manuscript fully available (please refer to the Data Availability Statement at the start of the manuscript PDF file)?

The PLOS Data policy requires authors to make all data underlying the findings described in their manuscript fully available without restriction, with rare exception. The data should be provided as part of the manuscript or its supporting information, or deposited to a public repository. For example, in addition to summary statistics, the data points behind means, medians and variance measures should be available. If there are restrictions on publicly sharing data—e.g. participant privacy or use of data from a third party—those must be specified.requires authors to make all data underlying the findings described in their manuscript fully available without restriction, with rare exception. The data should be provided as part of the manuscript or its supporting information, or deposited to a public repository. For example, in addition to summary statistics, the data points behind means, medians and variance measures should be available. If there are restrictions on publicly sharing data—e.g. participant privacy or use of data from a third party—those must be specified.

Reviewer #1: Yes

5. Is the manuscript presented in an intelligible fashion and written in standard English?

Reviewer #1: Yes

Reviewer #1: Most of the comments have been included as possible. Word limit needs to be checked to make it concise. Few grammatical checks required.

**Do you want your identity to be public for this peer review?** For information about this choice, including consent withdrawal, please see our Privacy Policy..

Reviewer #1: **Yes:** Dr Anushree Devashish Patil Scientist F (Deputy Director Senior Grade), Head of Division of Clinical Research, Indian Council of Medical Research, National Institute for Research in Reproductive and Child Health (ICMR-NIRRCH), MumbaiDr Anushree Devashish Patil Scientist F (Deputy Director Senior Grade), Head of Division of Clinical Research, Indian Council of Medical Research, National Institute for Research in Reproductive and Child Health (ICMR-NIRRCH), MumbaiDr Anushree Devashish Patil Scientist F (Deputy Director Senior Grade), Head of Division of Clinical Research, Indian Council of Medical Research, National Institute for Research in Reproductive and Child Health (ICMR-NIRRCH), MumbaiDr Anushree Devashish Patil Scientist F (Deputy Director Senior Grade), Head of Division of Clinical Research, Indian Council of Medical Research, National Institute for Research in Reproductive and Child Health (ICMR-NIRRCH), Mumbai

---

## [Editor Report · Decision Letter 2]

12 Mar 2026

A realist evaluation of the development, implementation and outcomes of the first public ART centre in Morocco

PGPH-D-25-01503R2

Dear Doctor Benbella,

We are pleased to inform you that your manuscript 'A realist evaluation of the development, implementation and outcomes of the first public ART centre in Morocco' has been provisionally accepted for publication in PLOS Global Public Health.

Best regards,

Julia Robinson

Executive Editor